# staRdom: Versatile Software for Analyzing Spectroscopic Data of Dissolved Organic Matter in R

**Matthias Pucher** [1,2,*] **, Urban Wünsch** [3] **, Gabriele Weigelhofer** [1,2] **, Kathleen Murphy** [3] **, Thomas Hein** [1,2] **and Daniel Graeber** [4]

[1]   WasserClusterLunz—Biologische Station GmbH, 3293 Lunz am See, Austria;
      gabriele.weigelhofer@wcl.ac.at (G.W.); thomas.hein@boku.ac.at (T.H.)
[2]   Institute of Hydrobiology and Aquatic Ecosystem Management, University of Natural Resources and Life
      Sciences, 1180 Vienna, Austria
[3]   Architecture and Civil Engineering, Water Environment Technology, Chalmers University of Technology,
      41296 Gothenburg, Sweden; wuensch@chalmers.se (U.W.); murphyk@chalmers.se (K.M.)
[4]   Department of Aquatic Ecosystem Analysis, Helmholtz-Zentrum für Umweltforschung—UFZ,
      39114 Magdeburg, Germany; daniel.graeber@ufz.de
*    Correspondence: matthias.pucher@wcl.ac.at; Tel.: +43-1-47654-81231

**Abstract:** The roles of dissolved organic matter (DOM) in microbial processes and nutrient cycles depend on its composition, which requires detailed measurements and analyses. We introduce a package for R, called staRdom ("spectroscopic analysis of DOM in R"), to analyze DOM spectroscopic data (absorbance and fluorescence), which is key to deliver fast insight into DOM composition of many samples. staRdom provides functions that standardize data preparation and analysis of spectroscopic data and are inspired by practical work. The user can perform blank subtraction, dilution correction, Raman normalization, scatter removal and interpolation, and fluorescence normalization. The software performs parallel factor analysis (PARAFAC) of excitation–emission matrices (EEMs), including peak picking of EEMs, and calculates fluorescence indices, absorbance indices, and absorbance slope indices from EEMs and absorbance spectra. A comparison between PARAFAC solutions by staRdom in R compared with drEEM in MATLAB showed nearly identical solutions for most datasets, although different convergence criteria are needed to obtain similar results and interpolation of missing data is important when working with staRdom. In conclusion, staRdom offers the opportunity for standardized multivariate decomposition of spectroscopic data without requiring software licensing fees and presuming only basic R knowledge.

**Keywords:** dissolved organic matter; DOM; PARAFAC; R; spectroscopy; fluorescence; absorbance; EEM; peak picking; drEEM; staRdom

## 1. Introduction

Dissolved organic matter (DOM) is the dominant organic matter form in nearly all aquatic ecosystems, where it modifies a plethora of ecosystem processes [1]. For example, it is a crucial nutrient source for microbial processes and a driver for microbial community composition in aquatic ecosystems [2–4]. DOM also controls different biogeochemical cycles, is a key modifier of primary and secondary production in lakes [5–7] and the ocean [8–11], and modifies microbial nitrate uptake [12,13] and changes the transport and the toxicity of pesticides in aquatic ecosystems (e.g., [14]). All these functions of DOM are related to its molecular composition. As significant fractions of DOM consist of chromophoric (light-absorbing) and fluorophoric (fluorescent) compounds [15,16], spectroscopic measurements can provide insights into DOM composition. Importantly, such ultraviolet–visible

spectroscopy is fast and cost-efficient, allowing immediate measurements during sampling campaigns with high sample throughput. Although some DOM fractions are not spectroscopically active and its spectral properties do not resolve fine details of its molecular structure, spectroscopic measurements can distinguish groups of DOM molecules in a meaningful way [17–19]. DOM spectroscopic properties have been shown to discriminate between sources, pathways, and undergone processes in aquatic ecosystems [20–24].

While absorbance data have mainly been used to estimate DOC concentrations and to detect the presence of aromatic compounds in DOM [25,26], fluorescence characteristics, such as the position, height, and shape of different fluorescence peaks, deliver additional information on DOM composition [15]. Electrons within DOM molecules are excited by photons of specific wavelengths and emit light at longer wavelengths due to energy loss (called Stokes shift; [27]). This Stokes shift defines the wavelength of fluorescence emission relative to the wavelength of excitation, and both depend on the structure of the excited DOM molecule. Three-dimensional excitation–emission matrices (EEMs) represent the spectroscopic measurements of DOM best [28], showing the complex picture of different, partially overlapping wavelengths of fluorescent light from various DOM molecules. Such EEMs cover a wide range of excitation and emission wavelengths (between approximately 200 and 700 nm) and can easily have more than 3000 data points per sample (e.g., [20]). For such large datasets, multivariate approaches are required to distinguish between chemically meaningful spectral patterns.

The most common approach for analyzing EEMs is Parallel Factor Analysis (PARAFAC, also called canonical decomposition, CANDECOMP) [29–33]. PARAFAC is a statistical model approach that extracts independent fluorophores from EEMs under ideal conditions [32]. Currently, the most popular software tools used to analyze EEMs by PARAFAC are the drEEM toolbox [32] and its predecessor DOMFluor [34], which both work in a MATLAB software environment. In both cases, the N-way toolbox calculates the actual PARAFAC model using an alternating least squares (ALS) algorithm [35]. Although these toolboxes are released under an open-source license, a commercial license for MATLAB is needed to run them.

Due to its open-source nature, the R software environment for statistical computing and graphics [36] is widely used in ecological research nowadays [37]. In R, a reliable PARAFAC toolbox is the multiway package [38]. The multiway package uses an ALS algorithm similar to N-way to solve PARAFAC models; however, the initialization procedure is limited in comparison to the ones offered by the N-way toolbox. Like N-way, multiway is not tailored toward a particular use case and thus requires in-depth knowledge of the R language for its application to spectroscopic data. While drEEM and DOMFluor have addressed this shortcoming, there is no spectroscopy-tailored PARAFAC toolbox available in R.

Besides PARAFAC analysis, many studies investigate specific absorbance spectra and fluorescence peak heights for analyses of DOM spectroscopic data. The method of "peak-picking" uses fluorescence intensities at predefined wavelengths or wavelength regions as a proxy for DOM composition [39]. In the aquatic sciences, commonly reported peaks include B (tyrosine-like), T (tryptophan-like), A (humic-like), M (marine humic-like), and C (humic-like) [40]. Furthermore, multiple indices are used to characterize aquatic DOM. Fluorescence-based indices usually use ratios of fluorescence intensities at two different predefined wavelengths or wavelength ranges, such as the humification index HIX, used to represent molecular complexity [41], the fluorescence index FI, indicating degree of autochthony [22], and the biological index BIX, showing recent autochthonous contributions [42]. Indices also exist for absorbance spectra, such as the specific UV absorbance at 254 nm (SUVA254) which indicates degree of aromaticity [25,43], and spectral slopes and slope ratios, which may relate to molecular size [44].

Data preparation is always necessary prior to PARAFAC modeling or index calculation (Figure 1). For fluorescence, this includes correction of wave dependency of the exciting light, dilution correction, inner-filter effect correction, signal normalization (Raman units or Quinine-sulfate equivalents), removal of Rayleigh scatter and Raman scatter, and outlier checks [32,45,46]. Currently available toolboxes that facilitate data preparation require either a commercial software environment (drEEM requires

MATLAB) [32] or a license themselves (Solo/MIA). For R, there is a free package available that can pre-process spectral data (eemR [47]). Until now, this package was not integrated into any PARAFAC package within R.

The need for a user-friendly software to correct and analyze EEMs was stated earlier [45]. We additionally identified the need for free software in a programming environment that ecologists and biogeochemists are familiar with [37] to reduce the initial hurdle of applying advanced data analysis methods. In this paper, we present and test a new package for fast and comprehensive spectroscopic analysis of DOM in R, called staRdom ("spectroscopic analysis of DOM in R"), which is suitable for the demands of both beginners and experienced users. This package combines and extends existing R packages, namely the multiway package [38] for PARAFAC and the eemR package [47] for data preparation and index calculation and provides additional ways for EEM data preparation and absorbance data analysis. Furthermore, staRdom includes plots and statistics based on established procedures for generating and validating PARAFAC models [32]. Due to its link to the eemR package, staRdom imports ASCII files generated with many fluorescence spectrometers, and accepts generic input files from less common instruments. staRdom delivers various output formats, including an option to transfer data to the OpenFluor database for existing PARAFAC models from DOM EEMs [48]. This study presents the capabilities of staRdom to analyze the optical properties of DOM and compares PARAFAC model results and the performance between staRdom (which uses the multiway package for model fitting) and drEEM (which uses the N-way toolbox for model fitting).

## 2. Materials and Methods

The staRdom package provides functions necessary for a detailed analysis of fluorescence and absorbance data of DOM, providing means for (i) data preparation, (ii) peak picking, calculations of (iii) fluorescence and (iv) absorbance indices, and (v) PARAFAC analysis. We show common approaches in the tutorial (Supplement S2); furthermore, specific functions are available for specific cases. For beginners, (i) data preparation, (ii) peak picking, (iii) fluorescence and (iv) absorbance index calculation can also be done using a template, which guides the user through the process. The user only needs to provide necessary data and parameters in an R Markdown (.rmd) file (plain text, refer to Supplement S1). No knowledge of R programming is necessary, and the calculations are automated. At the end of the analysis, staRdom places tables, plots, and a report of the analyses into a directory defined by the user. For (i)–(iv) and (v) PARAFAC analysis, all necessary steps and functions can be followed in the detailed tutorial (supplement S2). The user can either compile a specific R script with help from the tutorial or already use the example script and data for exercises and demonstrations.

Below, we introduce the basic features of staRdom including the respective commands of the presented functions. In the supplementary material, we provide a detailed manual for the version 1.0.26 of staRdom. Frequently updated, stable versions of staRdom together with vignettes for up-to-date tutorials and manuals can be downloaded from the staRdom pages on Comprehensive R Archive Network (CRAN, https://cran.r-project.org/package=staRdom) and newer, experimental versions can be downloaded from GitHub (https://github.com/MatthiasPucher/staRdom).

The offered functions follow the established concept [32]:

1.  Data preparation [47],
2.  Decomposing EEMs via PARAFAC/CANDECOMP [31,38], and
3.  Validating the model using a split-half analysis [49], the model fit and visual examination of the residuals.

In addition, staRdom includes the following features:

1.  Calculating fluorescence peaks and indices (tutorials S1 and S2) [47],

    ○    autochthonous productivity index/freshness index (BIX) [33,42],
    ○    classical peaks based on manual peak picking (B, T, A, M, C) [40],

○ fluorescence index (FI) [22] and
○ humification index (HIX) [41].

2. Calculating common absorbance (slope) indices:

○ absorbance at 254 nm (a254) [43],
○ absorbance at 300 nm (a300) [50],
○ ratio of absorbance at 250 to 365 nm (E2:E3) [51],
○ ratio of absorbance at 465 to 665 nm (E4:E6) [52],
○ spectral slope within log-transformed absorption spectra range ($S_{275-295}$, $S_{350-400}$, $S_{300-700}$) and the ratio of $S_{275-295}$ to $S_{350-400}$ (SR) [44],
○ the wavelength distribution of absorption spectral slopes [53] and
○ user-defined values and slopes can be extracted or calculated from the absorbance spectra.

In addition to these functions, staRdom provides experimental analysis approaches to stimulate research and offers scientists the opportunity to apply methods that have not yet been investigated in peer-reviewed publications. For example, staRdom allows recombining PARAFAC components from different models to project sample data onto a recombined model. We marked such functions as experimental in the documentation and the tutorial. Whenever possible, staRdom functions use multiple CPU cores to accelerate the calculation. This parallelization is used for simultaneously generated PARAFAC models with different random initializations as well as for the calculation of absorbance parameters and EEM interpolation of different samples.

For the demonstration, we used the example data from the drEEM tutorial [32] and followed the same analytical steps (Figure 1). The data consist of 206 fluorescence EEMs collected during seasonal surveys of San Francisco Bay [54]. Table 1 shows selected staRdom functions and the analytical step they are most probably used in.

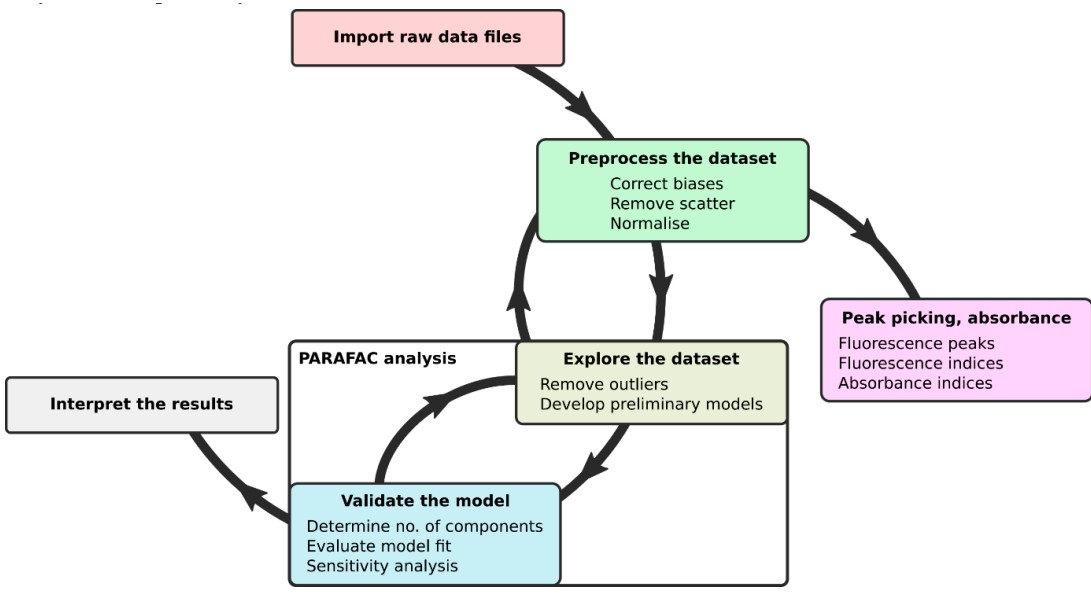

**Figure 1.** Scheme of Parallel Factor Analysis (PARAFAC) and index extraction within staRdom.

**Table 1.** List of selected staRdom functions and the step of the analysis scheme (Figure 1) that are most probably used.

| Step of the Analysis | staRdom Functions Used | Purpose of the Function |
|---|---|---|
| import raw data | *eem_read* | load EEM data |
| | *absorbance_read* | load absorbance data |
| check data consistency | *eem_checkdata* | check presence and names of samples |
| view data | *ggeem* | create single plots |
| | *eem_overview plot* | create multiple plots |
| correct biases | *abs_blcor* | absorbance baseline correction |
| | *eem_spectral_cor* | EEM spectral correction |
| | *eem_remove_blank* | subtract blank sample |
| | *eem_ife_correction* | inner-filter effect correction |
| | *eem_raman_normalisation,eem_raman_normali* | normalize EEM data to Raman units |
| remove scatter | *eem_rem_scat* | remove Rayleigh and Raman scattering of $1^{st}$ and $2^{nd}$ order |
| | *eem_setNA* | remove noise manually |
| | *eem_interp* | interpolate missing data |
| synchronize sample wavelength | *eem_red2smallest* | remove all wavelengths that are not present in all samples |
| | *eem_extend2largest* | create all wavelengths present in any sample in all samples |
| correct for sample dilution | *eem_dilution* | multiply EEM data by a dilution factor |
| smooth data | *eem_smooth* | smooth EEM data |
| normalize | no dedicated function, argument *normalise* = *TRUE* in *eem_parafac* | normalize EEM data |
| fluorescence peaks and indices | *eem_biological_index* | calculate BIX |
| | *eem_coble_peaks* | calculate Coble peaks |
| | *eem_fluorescence_index* | calculate FI |
| | *eem_humification_index* | calculate HIX |
| absorbance indices | *abs_parms* | calculate absorbance indices, spectral slopes, and selected ratios |
| calculate PARAFAC model (preliminary and final) | *eem_parafac* | calculate PARAFAC models |
| view PARAFAC models | *eempf_compare* | compare PARAFAC models (with different numbers of components) visually |
| | *eempf_comp_load_plot* | plot single PARAFAC models |
| identify outliers | *eempf_leverage* | calculate the leverage of each sample and wavelength |
| | *eempf_leverage_plot* | plot leverages |
| | *eempf_leverage_ident* | manually select samples in leverage plots |
| remove outliers | *eem_exclude* | remove samples and wavelengths from the data set |
| evaluate model | *eempf_convergence* | extract convergence behavior of a model |
| | *eempf_cortable, eempf_corplot* | show correlation between components |
| | *eempf_corcondia* | calculate the core consistency |
| sensitivity analysis | *splithalf, splithalf_plot* | calculate and plot a split-half validation |
| interpret the results | *eempf4analysis* | export table with component loadings |
| | *eempf_report* | create an analysis report in html format |
| | *eempf_openfluor* | export data for openfluor.org |

*2.1. Data Import*

In the first step, users import files written by the fluorometer into staRdom. Currently, staRdom can read ASCII data from Cary Eclipse, Horiba Aqualog, Horiba Fluoromax-4, Shimadzu, and Hitachi F-7000 as well as raw CSV-files.

For absorbance data import, data has to be provided by tables in CSV or TXT format (*absorbance_read*). A separate table containing dilution factors, Raman areas, or photometer pathlengths for each sample can be loaded to provide case-specific information in further analysis.

The function *eem_checkdata* checks for potential problems with the input data, such as missing data in samples, duplicate or invalid sample names, wavelength mismatches, and missing samples in either absorbance or fluorescence data. This data check does not change the original data but points at questionable characteristics in the data set. This function can be applied at any time in the analysis to ensure that the data is still coherent.

*2.2. Data Preprocessing*

staRdom partially applies functions from eemR [47] and uses additionally developed functions to correct EEM data and reduce noise in multiple ways. Here, we follow the same procedures used by

drEEM as described in Reference [32]. In short, the data are corrected for instrument-specific errors, making spectra comparable among measurements on different instruments and at different time points. Spectral correction factors are used to remove effects of wavelength-dependent shifts of the light source or the emission detector (*eem_spectral_cor*). Raman normalization (*eem_raman_normalisation*) is applied to remove instrument-dependent fluorescence intensity factors, such as different light source type, voltage, and light source age [45]. To produce linearity between sample fluorophore concentration and its fluorescence, the effect of internal light absorbance of the sample, i.e., the inner-filter effect, is removed (*eem_ife_correction*). To reduce or remove Rayleigh and Raman scatter a blank can be subtracted (*eem_remove_blank)* or the known area of the scattering can be removed specifically (*eem_rem_scat*). In addition, users can manually remove EEM data anywhere in the EEMs (*eem_setNA*). This may prove useful if noise has been introduced by instrument errors and parts of EEM spectra should be removed.

Missing values produced by scatter or other EEM data removal can be interpolated with different interpolation options (*eem_interp*). The default method used in staRdom is the multilevel B-spline approximation [55]. It is an accurate method to interpolate smooth but complex surfaces, which is exactly, what one would expect from a theoretical EEM. We suggest interpolating missing data whenever possible since the occurrence of non-converging models and models that converge in local minima has been shown to increase along with the proportion of missing values [56]. Therefore, *eem_checkdata* explicitly calculates the ratio of missing to nonmissing data to warn the user of this potential risk. To ensure a convenient interpolation in many cases, staRdom offers five different methods for interpolation from which the user can choose. The available methods are filling missing values with zeros, spline interpolation [55], piecewise cubic hermitean interpolation polynomials in one and two dimensions [57] and linear interpolation. In case of an analysis covering samples of different origin, samples measured with different instrument settings or array detectors (as used in the Horiba Aqualog), wavelength pairs in the sample set must be synchronized. Synchronization can be done by either reducing the data to wavelength pairs available in the whole set (*eem_red2smallest*) or adding and interpolating data that is missing in some samples but present in others (*eem_extend2largest*). Optionally, EEM data can be smoothed to stabilize results for calculations of fluorescence-based indices (*eem_smooth*). However, we do not recommend smoothing fluorescence EEM data before applying PARAFAC. Finally, the dilution of samples can be reverted by applying a dilution factor, where a factor of 2 represents a 1:2 dilution (*eem_dilution*).

We applied the following corrections to the example data: spectral correction, blank correction, inner-filter effect correction (Figure 2); and subsequently Raman normalization, scatter removal, wavelength cutting to remove spectral parts with low information content and high instrument noise, and interpolation of cut parts (Figure 3).

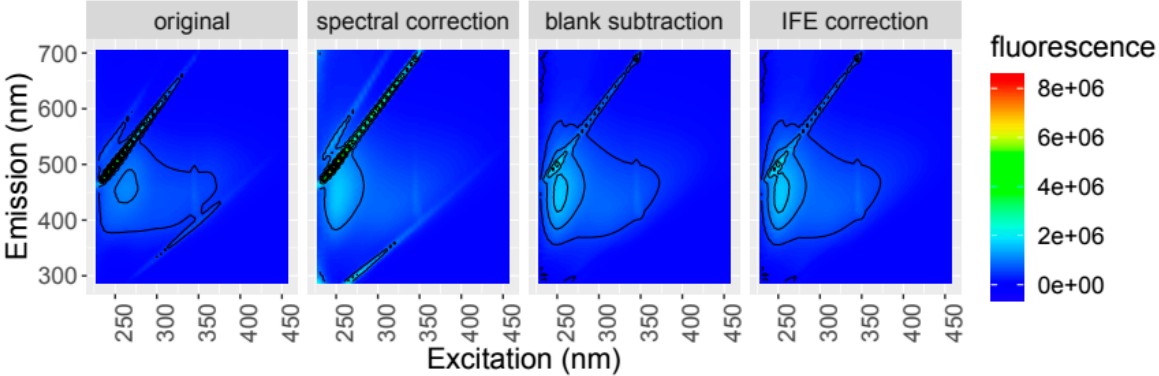

**Figure 2.** An excitation–emission matrices (EEM) sample as measured, untreated (original) and after spectral correction with correction factors, blank subtraction with ultra-pure water, and inner-filter effect correction using absorbance data.

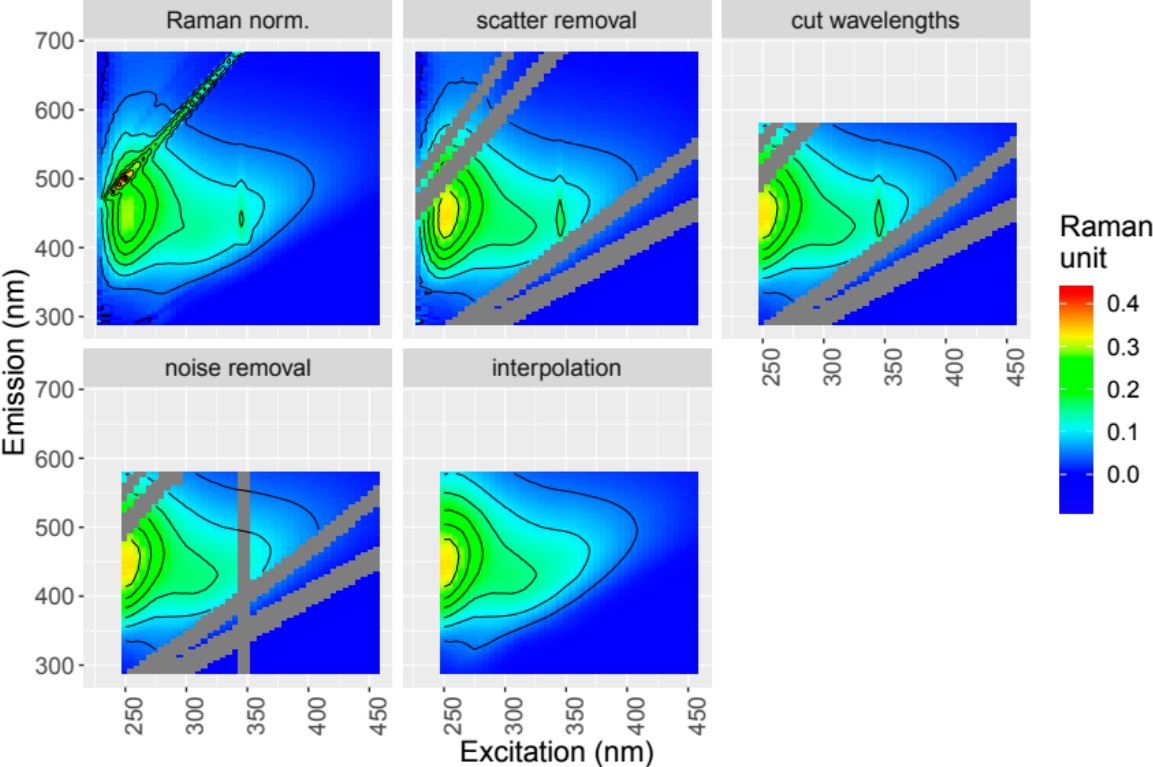

**Figure 3.** The same EEM sample as in Figure 2 after Raman normalization, scatter removal (Rayleigh 1st and 2nd order, Raman 1st and 2nd order), wavelength range reduction, manual noise removal, and interpolation of missing data.

We have also included absorbance data pre-processing steps in staRdom. Absorbance can be corrected by subtracting the mean of a chosen wavelength range (*abs_blcor*) [58] to correct a baseline drift in the instrument or correct for scatter due to particles. Within the chosen wavelength range the absorbance should be indistinguishable from the solvent (ultrapure water), which is usually the case at higher wavelengths (e.g., > 650 nm) when the measurement pathlength is small (<10 cm). Whether the water baseline is shifted from zero can be checked by comparing samples to ultrapure water measured right before or after the samples in question. Some instruments apply a baseline correction before the data is shown or exported and it does not have to be done by the user explicitly.

*2.3. PARAFAC Analysis*

2.3.1. Calculation of a PARAFAC Model from EEM Data

PARAFAC models assume that each sample set can be decomposed depending on a predefined number of components (N) using the following Equation (1) [29,30].

$$x_{ijk} = \sum_{f=1}^{N} a_{if} b_{jf} c_{kf} + e_{ijk} \tag{1}$$

In Equation (1), $x_{ijk}$ is the value of the $i$th sample, the $j$th emission wavelength, and the $k$th excitation wavelength. Here, $e_{ijk}$ is the respective residuum, i.e., data not modeled by PARAFAC components. $a$ (samples), $b$ (emission), and $c$ (excitation) are matrices (also called modes) with $N$ columns and multiple rows, equal to the numbers of samples, emission wavelengths, or excitation wavelengths, respectively. The matrices resulting from a multiplication of the vectors $b_f$ and $c_f$ show the PARAFAC components, resemble EEMs, and can, therefore, be interpreted easily.

A PARAFAC model can be calculated using *eem_parafac*. At a minimum, users need to supply the EEM data and the number of components N. For each N, a model is calculated and the comparison of peak shapes and model errors between multiple models can help finding the best solution. Optionally, the data can be normalized, which can be helpful if the samples span a wide range of DOC concentrations or if the components are highly correlated (see below). The model can be constrained to non-negativity, smoothness, or unimodality for each mode. Since fluorescence is always positive, non-negativity is commonly assumed for all modes [59]. The known shape of a fluorescence peak is smooth in emission and excitation modes and unimodal in the emission mode. Unimodality constraints are not always used because PARAFAC models typically show plausible results using only non-negativity constraints (e.g., [20,32]). Still, imposed unimodality can improve the interpretability of the results (e.g., [60,61]). The PARAFAC algorithm uses randomly generated start values. The stepwise least-squares approximation can return invalid local minima of the sum-of-squared-error (SSE) instead of the global minimum. To ensure that the global minimum is identified, several models are calculated using different random start values. The convergence tolerance can be set to provide an adjustable trade-off between accuracy and speed.

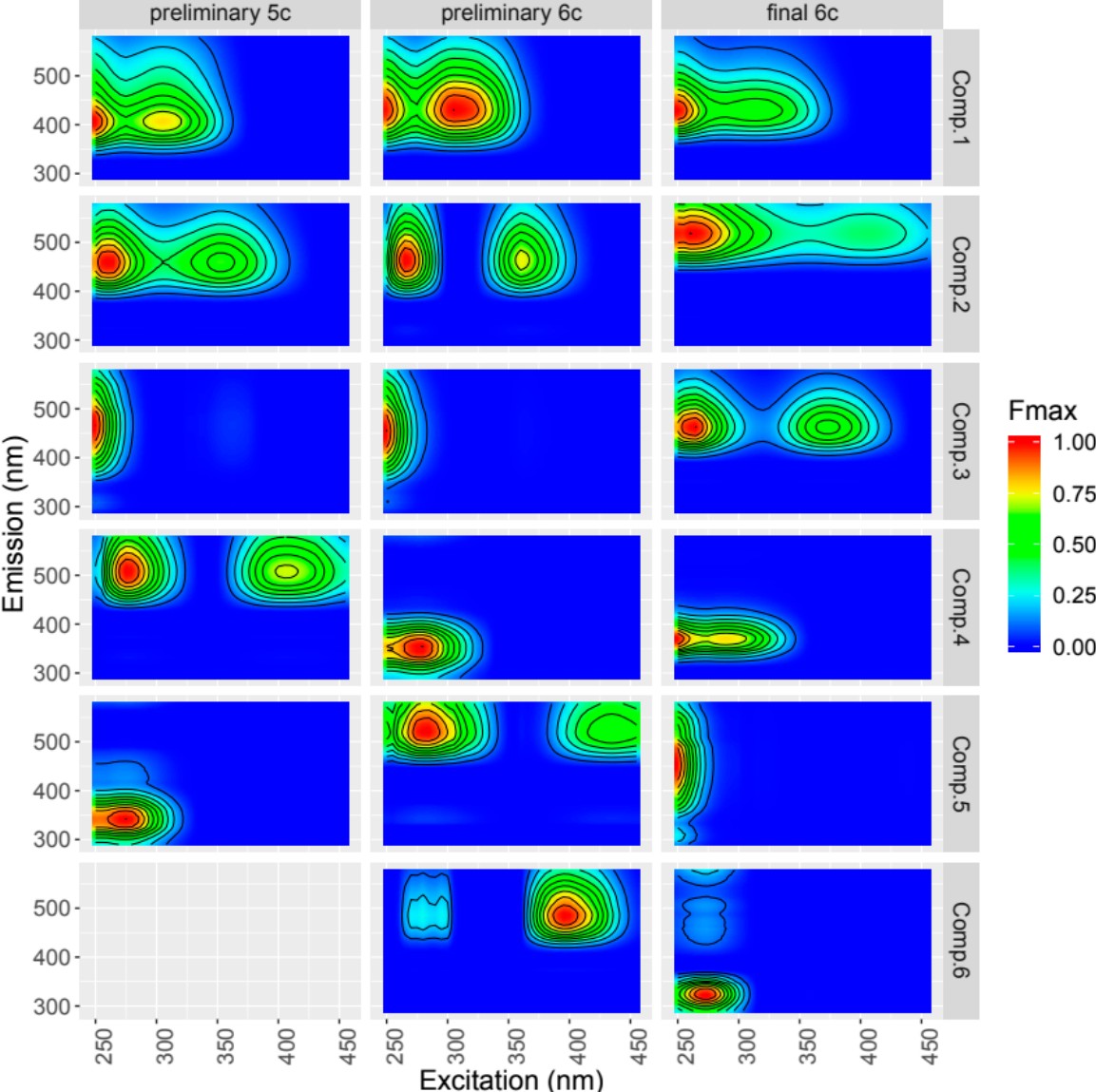

**Figure 4.** Comparing preliminary (outliers still included) and final models with 5 and 6 components. The components were normalized according to their maximum fluorescence (Fmax). Comp. = Component.

Plot functions can show the goodness of fit for the simultaneously calculated models and the components in two views, via matrix plots (Figure 4) and modes plots (not shown). Both are created using the function *eempf_compare*. These plots provide a quick overview on how well the model fits the data and how physically plausible the components are. Besides, different models can be compared visually. Studies describing shape characteristics of plausible fluorescence peaks [62] and peak shapes of pure substances can also be used to check for plausible shapes [63]. By plotting, different models can be compared visually. Certain model modes and components can also be plotted individually (*eempf_comp_load_plot*).

### 2.3.2. Identification of Outliers

In EEM datasets, outliers can result from sampling or analysis errors and can be identified if certain samples differ from others and influence the model [64]. Additionally, wavelengths in the emission or excitation modes can be identified as outliers, if there is noise at specific wavelengths in the majority of samples. Prior to a PARAFAC analysis, the best suitable data set needs to be found by analyzing both the noise and the influence certain samples or wavelengths have on the model. Decisions about removal of potential outliers are done via case-wise checks, where a new PARAFAC model is created with the reduced data set and compared to the original one.

In staRdom, outliers can be determined by calculating the leverage of each sample and each emission or excitation wavelength in a PARAFAC model (*eempf_leverage*). Values between 0 and 1 show the influence of each sample and each wavelength on the overall model. A function is available to identify samples and wavelengths visually, which should be considered for removal (*eempf_leverage_plot*, *eempf_leverage_ident*, Figure 5). Alternatively, the leverage can be viewed as a table (*eempf_leverage*, [31]). In the example, the samples dsfb676psp and dsgb447wt are identified as outliers because of a leverage much higher than all others.

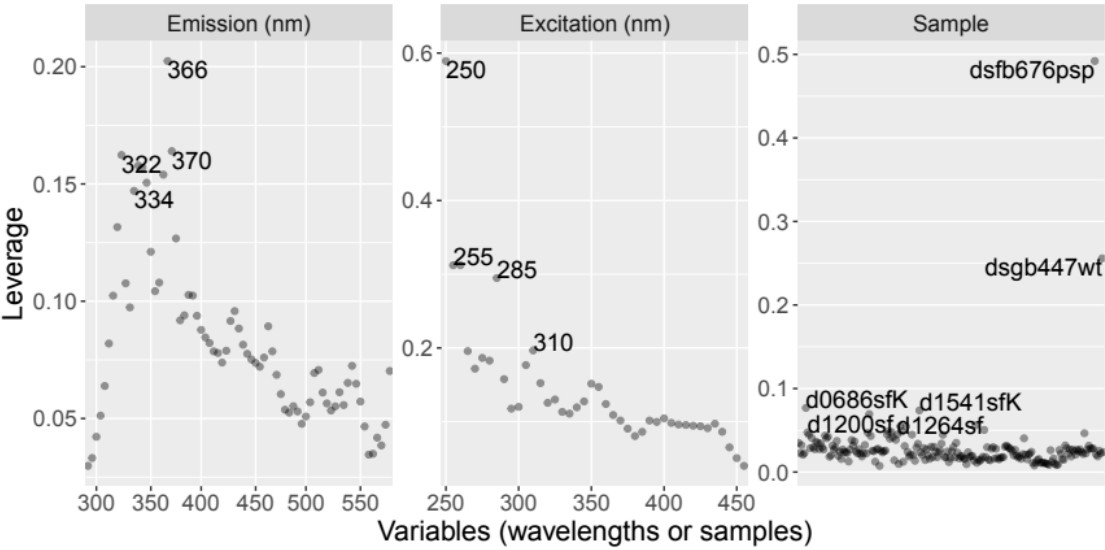

**Figure 5.** Outlier identification in a PARAFAC model using the leverage of emission and excitation wavelengths as well as samples.

Viewing the residual plots (*eempf_residuals_plot*) can also reveal unsuitable models. Residuals should only show random noise, which is especially common in areas where scattering was removed (Figure 6, samples A to D). Patterns can be easily seen and samples showing nonrandom data, maybe even peaks (Figure 6, samples E and F), can be identified.

If samples were removed before fitting the model, they can be reincluded later to see their loadings and residuals regarding the model (*A_missing*). This can help in interpreting outliers and including them in the analysis again despite their exclusion from the model creation. In Figure 6, the samples E

and F were considered outliers and removed before the model was created. Their residual plots show the part of the sample not covered by the final model.

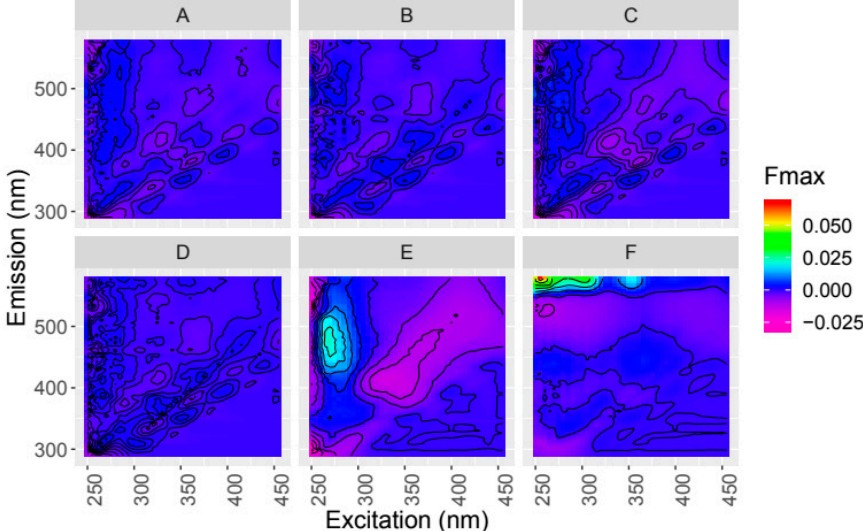

**Figure 6.** Residuals of 6 samples; E and F were identified as outliers using the leverage and, thus, were excluded from the model calculation.

### 2.3.3. Model Evaluation

Because the PARAFAC decomposition depends on the number of components N and this value is chosen by the user, finding an appropriate number N is an important task during the analysis. Hence, generating PARAFAC models is an iterative process where multiple models are compared and validated. These models can differ in terms of N, preprocessing, removed outliers, normalization of EEM matrices, and many more. Murphy et at. provide a detailed description on how to find a good PARAFAC model [32].

Depending on the data and the preprocessing steps, PARAFAC models can be unstable. Unstable models tend to show diverging results in different initializations either in convergence behavior or model error. The user can keep all of the calculated models and compare their results using the argument *output = "all"* in *eem_parafac* and extract the information using *eempf_convergence*. In case of less than, e.g., 5 similar, converging models, we advise to review the parameters used for model creation (e.g., increase the number of initialization, decrease the convergence criterion) and the preprocessing (e.g., interpolate missing data, use another interpolation algorithm).

For a solid interpretation of a PARAFAC model, the shape of the derived compounds should look similar to fluorescence patterns found in pure substances or combinations of pure substances [62,63]. This means they should be smooth in the excitation mode and smooth and unimodal in the emission mode [32].

The PARAFAC model assumes that the chemical species involved in the decomposition vary independently. There is an easy way to plot the correlation of components' loadings (*eempf_cortable*, *eempf_corplot*, Figure 7). This plot shows the distribution of the loadings for each component of the PARAFAC model diagonally as well as the regression curve (lower triangle) and the Pearson correlation coefficient (upper triangle) between two components. For datasets encompassing large concentration gradients, fluorophores can strongly across the dataset, challenging the assumption of independent variability of the PARAFAC components. Furthermore, very high correlation between component scores can indicate over-fitting, i.e., too many components used in the PARAFAC model. One preliminary model in the example had highly positively correlated components 1 and 2 (Figure 7a). After normalization of the EEM data over each sample, the collinearity is greatly reduced (Figure 7b).

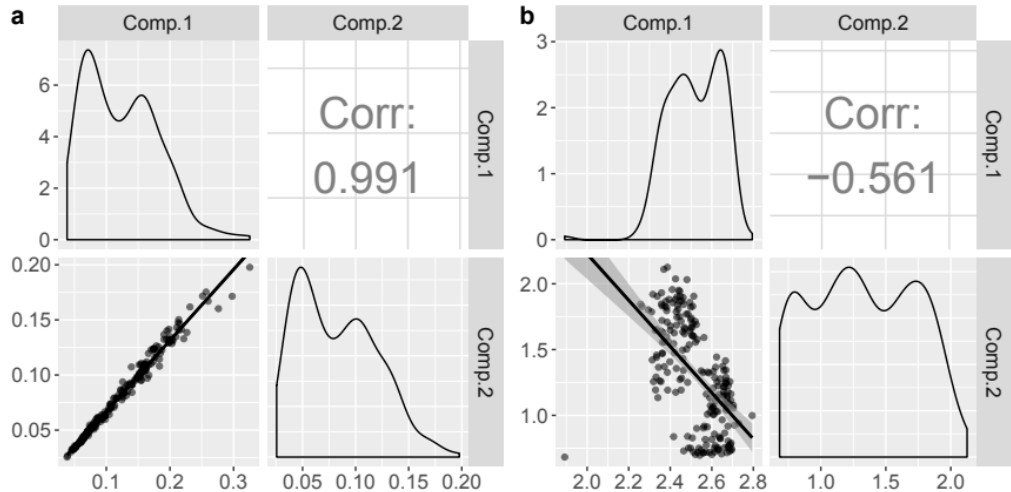

**Figure 7.** Components' loadings from PARAFAC models (**a**) without and (**b**) with previous normalization. Diagonally: distributions of the loadings; lower triangle: regression curve between components; and upper triangle: Pearson correlation coefficient between components.

The models are validated using a split-half validation, which is a specific method of cross-validation [32]. A robust model validation requires enough samples (depending on the data approximately 100–200 samples). Split-half validation is realized with the *splithalf* function in staRdom. Here, models are compared based on different subsets of the original data. The implemented test uses 4 splits, 6 combinations, and 3 tests ($S_4C_6T_3$): the data are split into four subsets (A, B, C, and D) and recombined to compare one half of the data to the other in different combinations (AB–CD, AD–BC, AC–BD) [32]. The comparison is done visually by plots showing the spectral loadings (*splithalf_plot*, Figure 8) and by calculating Tucker's congruence coefficient [65] (TCC) or shift- and shape-sensitive congruence [66] (SSC, *splithalf_tcc*). Subsets can be automatically generated or manually defined.

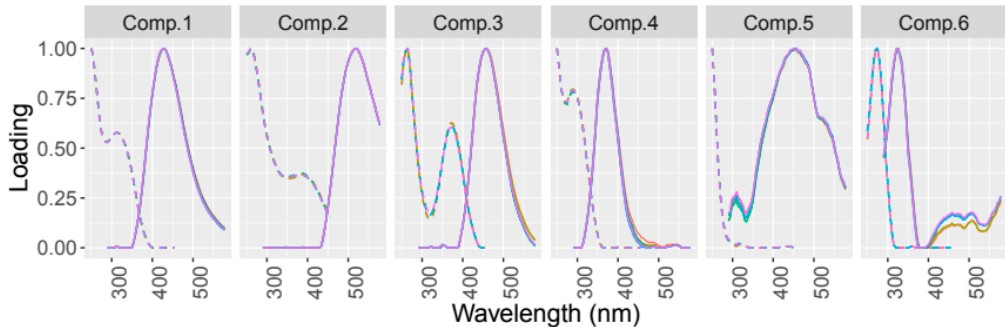

**Figure 8.** Split-half validation of a PARAFAC model. Emission spectra: full line; excitation spectra: dashed line; colors according to models using different sample sub-sets.

The quality of a PARAFAC model can also be tested using the core consistency (*eempf_corcondia*) [67]. Models specified with an appropriate number of components should have high core consistencies (near 100%), whereas low core consistencies indicate that too many components were specified. The core consistency diagnostic should protect against over-fitting but rigid adherence to guidelines can lead to under-fitted models. For real-life fluorescence datasets, it has been suggested that the core consistency diagnostic is overly severe [31]. Further research is needed to establish the utility of the core-consistency diagnostic for validating PARAFAC models of natural organic matter EEMs.

### 2.4. Export and Further Interpretation of Results

For further analysis and interpretation, staRdom offers a table containing the loadings of each component per sample (*eempf4analysis*). Peaks and indices from EEM data can be included in this table.

PARAFAC models can be exported to a file and uploaded to openfluor.org (*eempf_openfluor*) [48]. By doing that, a comparison to published and partly peer-reviewed models is possible and can help in interpreting the results.

An overview of a model optionally including information, such as applied correction methods, leverages and split-half validation, can be written as an HTML file (*eempf_report*).

*2.5. Toolbox Comparison*

We compared the results from multiple PARAFAC models using staRdom (PARAFAC model calculated by the multiway package) and drEEM (PARAFAC model calculated by the N-way toolbox). The models were derived from four published datasets, which comprised of marine, lake, and stream samples from different areas and climate zones and were measured on different instruments from different ecosystems and/or from different landscapes. Additionally, two datasets consisting of pure fluorophore spectra were compared (Table 2).

For each dataset, 3000 PARAFAC models were calculated with convergence criteria varying between $1 \times 10^{-6}$ and $1 \times 10^{-9}$ for drEEM and between $1 \times 10^{-8}$ and $1 \times 10^{-11}$ for staRdom, in steps of $1 \times 10^{-1}$. Different convergence criteria were necessary because the toolboxes use different methods to monitor convergence, as shown in Equations (2) and (3). Due to this difference, when identical convergence criteria are supplied in the function inputs to both staRdom and drEEM, then drEEM return models with a smaller modeling error.

staRdom:

$$\frac{(SSE_n - SSE_{n-1})}{\sum X^2} \leq crit \tag{2}$$

drEEM:

$$\frac{(SSE_n - SSE_{n-1})}{SSE_n} \leq crit \tag{3}$$

$SSE_n$—sum-of-squared-error of nth iteration
$X$—EEM data
*crit*—convergence criterion

For both toolboxes, the maximum number of iterations was set to 2500, and non-negativity constraints were applied in all modes. The time until convergence (TUC) was measured by initializing a timer function just before each call to the PARAFAC function and stopping it just after completion of each call to the respective PARAFAC functions. The remaining model metrics were supplied by the respective toolboxes and included the number of iterations until convergence, and the sum-of-squared-error (SSE) of each model. Lastly, the number of models that reached the iteration limit or stopped due to other reasons before convergence was compared.

To show the influence of the number of random initializations we used a Monte-Carlo simulation, i.e., the respective number of models was picked from the whole set of models 5000 times randomly for each data set and convergence criterion. The sum of TUC of the models in a subset was considered to equal the calculation time of a set under realistic conditions running on a single CPU core. The best model per subset was used as a representative for this set, to mimic a standard analysis. From within all models of each data set, the one with the least SSE was used as a reference for model quality. We calculated the TCC to compare each of them with the best models [48]. The TCC is a parameter for assessing similarity between pairs of fluorescence excitation and emission spectra and ranges between 0 (totally different) to 1 (identical). The 99% quantile of SSE within all model subsets with the same convergence criterion and the same number of initializations was considered being the accuracy that can be achieved using exactly these conditions. Model parameters were accepted as sufficiently accurate if the TCCs of all components were at least 0.999. The TCC is a parameter for model similarity between 0 (totally different) to 1 (identical). In comparisons between different studies a TCC of 0.95 is considered to show a good similarity between two components [48].

**Table 2.** Test datasets and important characteristics.

| Name | Number of | | | | % | Description | Reference |
|---|---|---|---|---|---|---|---|
| | Comps [1] | Samples | Em [2] | Ex [3] | NA [4] | | |
| Amino3 | 3 | 5 | 201 | 61 | 0 | Pure amino acids | [31] |
| Fjord6 | 6 | 191 | 91 | 44 | 16.6 | Solid-phase extracts of DOM from three arctic fjords | [68] |
| Headwater4 | 4 | 235 | 151 | 43 | 0 | Headwater streams, and agricultural catchments, Denmark and Uruguay | [20] |
| PortSurvey6 | 6 | 206 | 73 | 42 | 9.5 | port and oceanic marine samples (USA, Pacific coast), drEEM tutorial dataset | [54] |
| Pure5 | 5 | 60 | 50 | 40 | 0 | Pure substances with added artificial noise | unpublished |
| RioEx4 | 4 | 58 | 97 | 111 | 0 | Photodegradation experiment of solid-phase extracted DOM | [69] |

[1] components of the PARAFAC model, [2] emission wavelengths, [3] excitation wavelengths, [4] missing data.

## 3. Results and Discussion

For five of the six tested datasets, PARAFAC models obtained from both toolboxes were highly similar to the best solution with TCCs of at least 0.999 in scores and loadings. For one of the datasets (Fjord6), no convergent models were obtained using staRdom; possible reasons and solutions for this are missing data (see Section 3.3). Both software tools required different considerations regarding model parameters (Figure 9 and Table 3). As such, we identified the convergence criterion, the number of random initializations and susceptibility toward missing data as important factors during the comparison of staRdom and drEEM. The impacts of different model parameters on the results are highly dependent on the data set. In the following, we address particularities and provide suggestions for smooth practical work and mitigations of possible problems.

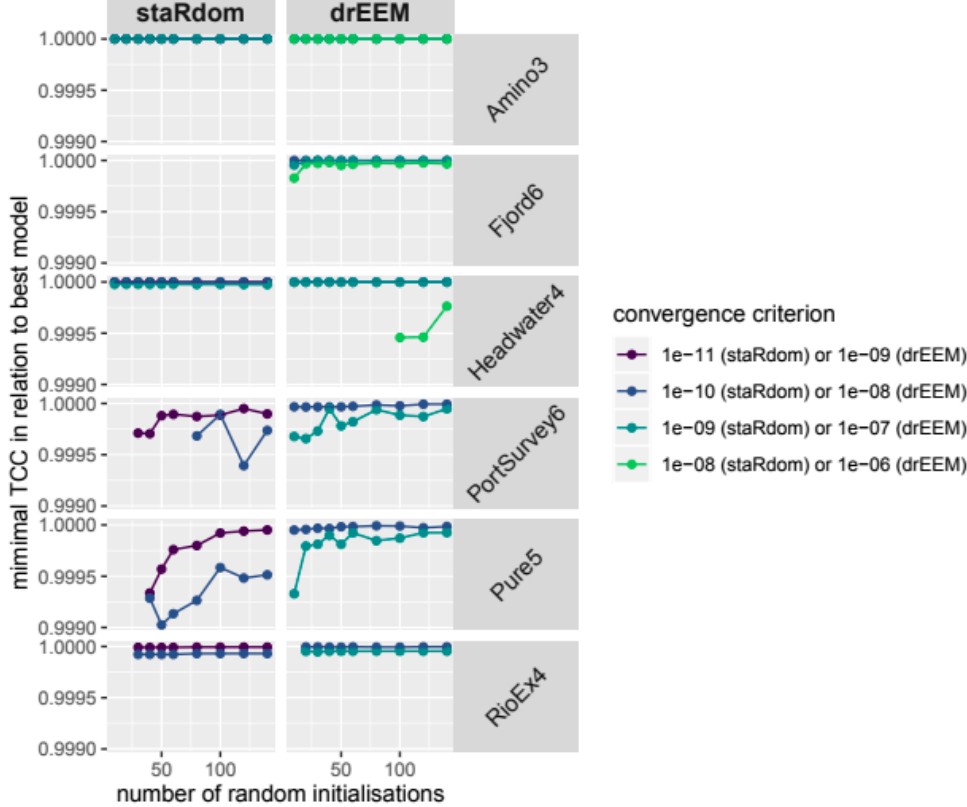

**Figure 9.** Similarity of PARAFAC models derived using staRdom and drEEM, using different numbers of random initialisations and different convergence criteria. Similarity is measured as minimum TCC: the higher the TCC, the more similar are the components.

**Table 3.** Summary of test performance for five datasets that produced convergent PARAFAC models.

| Name | Software | Convergence Criterion | Initializations | Relative Error | Minimum TCC |
|---|---|---|---|---|---|
| Amino3 | staRdom | $1 \times 10^{-8}$ | 10 | 1.000069 | 1.0000 |
| | drEEM | $1 \times 10^{-8}$ | 10 | 1.000000 | 1.0000 |
| Headwater4 | staRdom | $1 \times 10^{-9}$ | 10 | 1.000010 | 1.0000 |
| | drEEM | $1 \times 10^{-7}$ | 10 | 1.000002 | 1.0000 |
| PortSurvey6 | staRdom | $1 \times 10^{-11}$ | 30 | 1.000042 | 0.9997 |
| | drEEM | $1 \times 10^{-7}$ | 10 | 1.000039 | 0.9997 |
| Pure5 | staRdom | $1 \times 10^{-10}$ | 40 | 1.000071 | 0.9993 |
| | drEEM | $1 \times 10^{-7}$ | 10 | 1.000056 | 0.9993 |
| RioEx4 | staRdom | $1 \times 10^{-10}$ | 30 | 1.000022 | 0.9999 |
| | drEEM | $1 \times 10^{-7}$ | 20 | 1.000015 | 1.0000 |

### 3.1. Number of Initializations

Implementing repeated starts of PARAFAC models under identical conditions but with different (random) starting values is common practice to identify a robust least-squares solution. If an insufficient number of random initializations is used, a local minimum solution may be identified instead of the global least-squares solution.

Our analysis showed good results with staRdom for five datasets assuming 40 initializations (Figure 9). This indicates that it is preferable to start at least 40 models in order to obtain a solution that is sufficiently stable and close to the global minimum of the PARAFAC modeling error. Users should monitor the error of all solutions and increase the number of random starts, if the best solution is far better than the others.

However, an issue was sometimes encountered whereby a proportion of models did not converge. Except in the case of the Fjord6 dataset, multiple convergent solutions were obtained using staRdom and a reasonable model could be obtained from within the subset of models that converged. As a solution for slow and incomplete convergence, Reference [56] demonstrated the interpolation of missing data.

staRdom monitors the number of models that converged within the specified number of iterations. By adding the argument *output = "all"* in the function *eem_parafac*, *eempf_convergence* can provide detailed information on the convergence behavior of the model. As a general precaution, *eem_parafac* always informs the user if less than 50% of models converged. In response, users can either increase the number of random starts or specify that models should be calculated until a specified number of convergent models have been produced (*strictly_converging = TRUE* in *eem_parafac*). This function was not applied in the demonstration and the number of models shown in the results contain both convergent and nonconvergent models. drEEM does not currently provide a similar function for tracking nonconvergence, but nonconvergence appeared to occur less frequently analyzing the described data sets.

### 3.2. Convergence Criterion

For the datasets we investigated, staRdom provided a similar modeling error as drEEM as long as the convergence criterion was increased by two to three orders of magnitude (Figure 9 and Table 3). Therefore, the default convergence criteria are $1 \times 10^{-6}$ in drEEM and $1 \times 10^{-8}$ in staRdom. As the results in this study are based on model errors, these differences do not further influence any results of the shown PARAFAC models.

### 3.3. Influence of Missing Data

Only two of the six test datasets (Fjord6 and PortSurvey6) contained missing data corresponding to regions of Rayleigh and Raman scatter, while the remaining datasets were interpolated prior to PARAFAC modeling. It seems likely that for the Fjord6 dataset, a relatively large proportion of missing numbers (16.6% missing) was a causal or contributing factor explaining why staRdom did not reach

convergence prior to the maximum number of iterations. It appears that staRdom may be more sensitive to missing data than drEEM, although further tests are needed to determine if this is the case.

Previous studies have shown that missing data should generally be interpolated in order to avoid local minima [56]. In order to obtain robust results, we stress the advantage of interpolating areas of Rayleigh and Raman scatter. To support users in finding an interpolation leading to a reasonable PARAFAC model, staRdom offers five different interpolation methods (see Section 2.2). Future studies and developments in the multiway package should address this issue to improve the convergence behavior where interpolation is no option for mitigation.

*3.4. Time until Model Convergence*

Since PARAFAC modeling of large datasets can be time-consuming, the time elapsed until model convergence (using parameters as stated in Table 3) using three common CPU architectures (introduced 2013–2017) was compared. For both toolboxes, the algorithms reached convergence within similar timespans, even in cases where staRdom calculated with a higher number of initializations (Figure A1). Comparing time elapsed for the oldest to the newest CPU model gives an idea of how modeling speed is affected by improvements in hardware. For the computers tested in this study, improvements in CPU reduced the time until convergence by approximately 50% in MATLAB and 20% in R.

*3.5. Outlier Calculation and Split-Half Validation*

Both, the outlier identification [31] and the split-half analysis [49] rely on a PARAFAC model. The approaches applied after the PARAFAC algorithm are implemented identically in staRdom and drEEM, so we limited our tests on the PortSurvey6 dataset only.

For the PortSurvey6 data, outlier identification (Figure 5) and split-half analyses (Figure 8) produced essentially identical outputs using either staRdom or drEEM.

## 4. Conclusions

We introduced "staRdom", a new toolbox for the analysis of absorbance spectra and fluorescence EEMs using the R statistical computing environment. Data preprocessing steps and routines in staRdom are for the most part identical to those available in the established drEEM toolbox for MATLAB. Results from both toolboxes are interchangeable apart from datasets with a relatively high fraction of missing data. The availability of multiway analysis tools in the free R software environment will reduce barriers in spectroscopic research and stimulate advances in DOM biogeochemistry for natural and engineered systems.

**Supplementary Materials:** The following are available online at http://www.mdpi.com/2073-4441/11/11/2366/s1, Document D1: Correcting raw data, calculating peaks and indices in EEMS and absorbance (slope) parameters, Document D2: PARAFAC analysis of EEM data to separate DOM components.

**Author Contributions:** Conceptualization, M.P. and D.G.; methodology, M.P., D.G., U.W., and K.M.; software, M.P.; validation, K.M., G.W., and T.H.; formal analysis, M.P., U.W., and K.M.; investigation, K.M. and D.G.; resources, G.W. and T.H.; data curation, M.P.; writing—original draft preparation, M.P.; writing—review and editing, M.P., G.W., K.M., U.W., T.H., and D.G.; visualization, M.P.; supervision, D.G. and T.H.; project administration, G.W.; funding acquisition, G.W.

**Funding:** This research was funded by the Provincial Government of Lower Austria, Nö Forschungs- und Bildungs GmbH, within the Science Call 2015 (SC15-002, project ORCA). K.R.M. and U.J.W. acknowledge funding from the Swedish Research Council (FORMAS 2017-00743). M.P. acknowledges funding from the doctoral school Human River Systems in the 21st century (HR21).

**Acknowledgments:** A very important part of this work was the feedback from users that were interested and dedicated at an early stage already. We would like to thank especially Stefan Preiner, Astrid Harjung, Renata Pinto, Ching-Hsuang Lo, Alexandra Tiefenbacher, and Gerardo Gold-Bouchot for testing, asking, and stimulating the development. Nora and Christoph Zechmeister contributed to the staRdom logo used in the tutorials.

**Conflicts of Interest:** The authors declare no conflict of interest. The funders had no role in the design of the study; in the collection, analyses, or interpretation of data; in the writing of the manuscript; or in the decision to publish the results.

# Appendix A

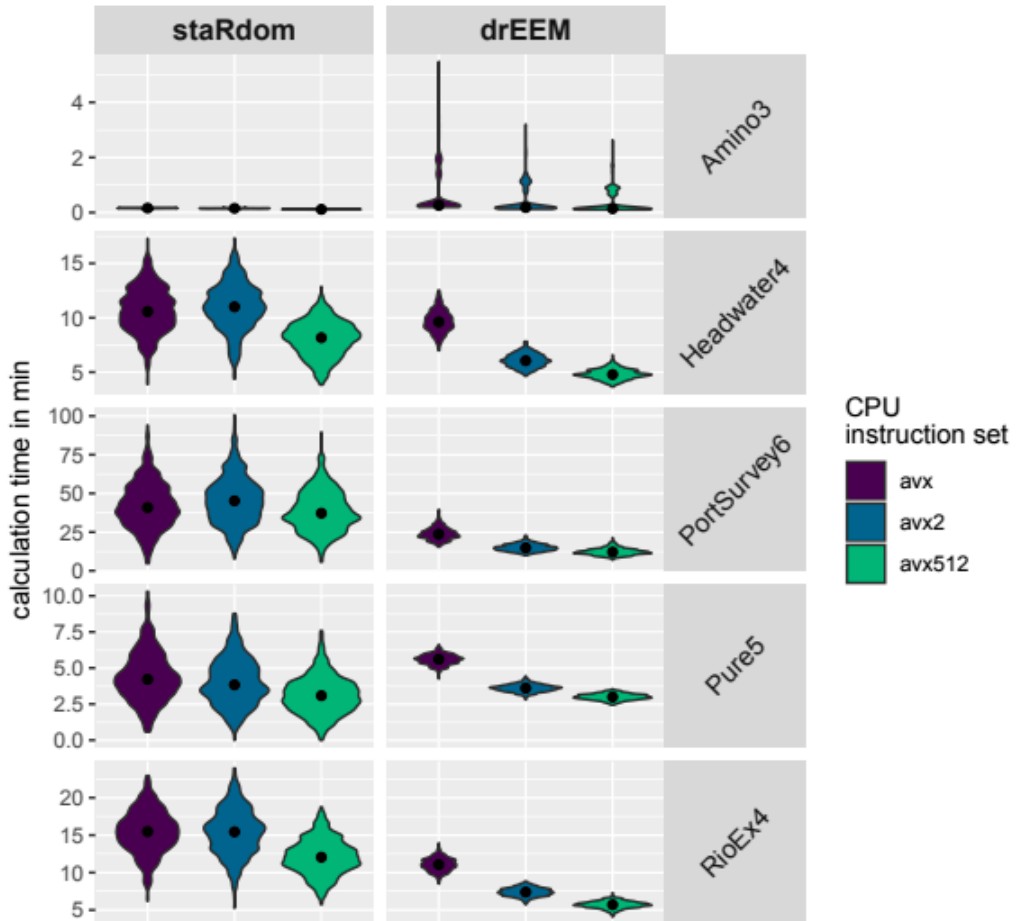

**Figure A1.** Distributions of calculation times for models of five datasets, using staRdom and drEEM on different CPU architectures (CPU speed improves from left to right), with the model specifications defined in Table 3.

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
