# Peer review of "staRdom: Versatile Software for Analyzing Spectroscopic Data of Dissolved Organic Matter in R"

_water, doi:10.3390/w11112366_

Round 1

Reviewer 1 Report

This manuscript presents, describes and evaluates a new, free software toolbox called "staRdom" to analyze optical data from dissolved organic matter (DOM) samples, associated with the free statistical computing environment, R. The authors are correct that the existing toolboxes, themselves, are freely available, but one must purchase a license to the related software (i.e., Matlab). Thus, a freely-available toolbox for analyzing excitation-emission matrices (EEMS) on a open software platform like R is needed and I agree that something like this would advance scientific investigation of DOM (and related biogeochemical processes). Overall, to me, this paper simply reviews the mathematics/statistics behind EEM processing and analysis, while promoting the new toolbox (thus, I did not rate the novelty and significance of content very highly). Also a result of this is that I have no substantial comments on the science/process (they have it right and these are well-known functions/analyses).

However, the intercomparison of staRdom to currently used PARAFAC toolboxes is a real strength of the paper. It indicates that staRdom results may be comparable to past drEEM results (without significant portions of data missing). Combined with the supplemental materials (including web materials), I see no obvious 'barriers to entry' for anyone new or accustomed to DOM optical analyses. Indeed, I plan to try this software on my next DOM dataset.

The paper is well written and has only minor formatting issues that I will leave for the editorial process. The real test of this software will be how it performs when in the hands of other researchers. 

Author Response

We would like to thank the reviewer for taking his or her time to read the manuscript and sharing his or her opinion on the manuscript.

Reviewer 2 Report

Dear Editor –

The manuscript titled “staRdom: versatile software for analyzing spectroscopic data of dissolved organic matter in R” by Pucher et al. releases a new coding packing for the open source statistical software R, to allow for open source parallel factor analysis (PARAFAC) of 3-D fluorescence spectra of dissolved organic matter (DOM). This is a massively impactful development for the scientific community, as currently there is no available PARAFAC analysis that utilizes open source software. PARAFAC analysis has grown significantly in use over the past decade, and while the code for PARAFAC has always been broadly available, it has relied upon use with expensive commercial software. Releasing a PARAFAC package in R will broadly expand the usage of this analysis. The authors have done an excellent job providing all of the statistical tools in a user-friendly manner to fully evaluate fluorescence and absorbance spectra of water samples through the entire process. This includes correction after acquisition to the determination of spectroscopic indices to the resolving of PARAFAC components, and is set up for data formats from several popular fluorimeters, with the flexibility for custom formats.

The authors did a thorough analysis of the convergence differences between DrEEM in MATLAB and staRdom in R, and showed through systematic comparison the similarity of results between the two packages. Also appreciated was the inclusion of the scalability analysis of staRdom. There are several included graphical versions of statistical analyses such as leverage, residuals, the multiple forms of split half analysis, and random initializations, and easy to apply features such as cropping out noise from EEMs. Overall, this is a much-anticipated development for the fluorescence community, and this package is likely to be highly utilized. A few minor comments and suggestions are listed below:

It is likely that scientists using this package may not be experts in PARAFAC yet and may benefit from a bit more provided guidance. Overall, it would be useful to provide a example set of files/dataset for the tutorial and expected output from sets of coding comments, to allow users to test that they are applying code correctly. This could include files such as a set of EEMs, example correction files, etc. and provided output. To that end, the discussion of interpolation options (lines 231-233) could be expanded to help a new user identify which interpolations might be appropriate, or reference these discussions elsewhere. These may be better discussions for the tutorial document rather than the manuscript itself. It is unclear why a threshold of 50% was chosen for the number of converged models (lines 519-520). This threshold should be referenced or the rationale for it further explained.

Author Response

We would like to thank the reviewer for taking his or her time to read the manuscript and sharing his or her valuable suggestions and questions on the manuscript.

Overall,... : The tutorial (supplement S2) explains the analysis of the drEEM example data including how to download and import the raw data. The expected output can be seen in the tutorial. For package size limitations on CRAN, only limited example data are included in the package itself. They allow to test simple correction functions and include the results from the PARAFAC analysis and further test, but to calculate the presented PARAFAC model itself, users have to download additional data using the provided R code.

lines 231-233: We added a short explanation about the default method to the manuscript: The default method used in staRdom is the multilevel B-spline approximation. It is an accurate method to interpolate smooth but complex surfaces, which is exactly, what one would expect from a theoretical EEM.To our knowledge, there is no study investigating the effects of different interpolation methods on the results of a PARAFAC analysis. The possibility of using different methods in staRdom could induce such an investigation.In the tutorial, we added a sentence, that unusual residuals or convergence behaviour might be improved by a different interpolation method. This was also our motivation for implementing the interpolation providing different methods.

lines 519-520: We added a more detailed description in the manuscript:By adding the argument output = “all” in the function eem_parafac, eempf_convergence can provide detailed information on the convergence behaviour of the model. As a general precaution, eem_parafac always informs the user if less than 50 % of models converged.

Reviewer 3 Report

In the present manuscript, the authors introduce a new package for R, called staRdom to analyze DOM spectroscopic data (absorbance and fluorescence). The comparison between both tools (stardom and drEEM) is convincing. This new toolbox appears to be a good alternative for fluorescence and absorbance data treatment for the non-Matlab and open-source addicts. The paper is well written. However, the manuscript needs a few improvements in order to clarify some parts. Therefore, I recommend the paper for publication after moderate revision.

General comment:

The M&M part is a bit dense. Beginners can be quickly lost. I do not suggest to reduce the text but help the reader to visualize the information. I would suggest two things:

i) Reorganize the subtitles of M&M according to Figure 1 for consistency. In my idea, the following plan will be more appropriate.

2.1 Data import

2.2 Data pre-processing

2.3 PARAFAC Analysis

2.3.1 Calculation of a PARAFAC model from EEM data

2.3.2 Identification of outliers

2.3.3 Model evaluation

2.4 Export and further interpretation of results

2.5 Toolbox comparison

ii) To help the user and specifically the beginners, it would be interesting to have a figure/table or a procedural scheme combining figure 1 with the used functions for each step.

Specific comments:

Line 36-41: This is a bit restrictive and details can be added.

Line 93-99: As you present HIX and FI indices I recommend to present BIX (Huguet et al., 2009).

Lines 134: Here, a reference for the tutorial is needed.

Line 440: Replace “stardom” with “staRdom”.

Line 479: Landscape orientation will be more appropriate for this table. In the current form, it is difficult to read the table header.

Line 484: Authors named the dataset “fjord6” or in Table 1 it is named “Fjord”. Please, be consistent.

Line 485: What data are “missing data”? Could you explain more the reason for non-convergence? Or refer to part 3.3.

Line 493: Again, use the same name between the figure 9 and the Table 1 and also the same order (e.g., Fig. 9 minimal TTC of Pure 5 should be presented after Rioex 4.

Line 498: Same comment for Table 2 than for Table 1.

Line 560: What about the other dataset?

Line 565: This sentence should be removed from the conclusion.

Author Response

We would like to thank the reviewer for taking his or her time to read the manuscript and sharing his or her valuable suggestions and questions on the manuscript.

M&M part: We followed the proposed section structure of the reviewer.

We added a table with a non-exhaustive list of staRdom functions and linked them to the scheme shown in figure 1.

Line 36-41: We agree that the examples were a bit restrictive, however, we just wanted to present those examples to underline the importance of a simple DOM composition analysis for many applications. We now add a sentence clarifying the multitude of modified aquatic processes and transport patterns and that we just present some examples for those. We do not want to lengthen this part, as the paper is on the spectroscopic data analysis toolbox not on DOM-related ecosystem processes.

Line 93-99: We followed the suggestion of the reviewer.

Lines 134: We followed the suggestion of the reviewer.

Line 440: We followed the suggestion of the reviewer.

Line 479: We agree with the reviewer, that the readability could be improved. In favour of readers using a PDF version of the manuscript, we do not want to introduce landscape tables. Therefore, the readability of the table was improved by shortening the table header and decreasing the font size.

Line 484: We changed the names of the datasets to be consistent over the whole document.

Line 485: Missing data stem from scatter removal. We refer to section 3.3 at that point. A detailed explanation of the convergence behaviour was not in the scope of this work and would need further test.

Line 493: We changed the names and orders of the datasets to be consistent over the whole document.

Line 498: We agree with the reviewer, that the readability could be improved. In favour of readers using a PDF version of the manuscript, we do not want to introduce landscape tables. Therefore, the readability of the table was improved by shortening the table header and decreasing the font size.

Line 560: We added a sentence on why the other datasets were not tested. Both, the outlier identification and the split-half analysis rely on a PARAFAC model. The further ideas to derive the leverage or a split-half analysis are implemented identically in staRdom and drEEM, so we limited our tests on the PortSurvey6 dataset only.

Line 565: We followed the suggestion of the reviewer.

Round 2

Reviewer 3 Report

All my comments have been well addressed.